# A dish-like molecular architecture for dynamic ultralong room-temperature phosphorescence through reversible guest accommodation

Wenlang Li[1,2,5], Qiuyi Huang[1,5], Zhu Mao[1], Xiaoyi He[1], Dongyu Ma[1], Juan Zhao[3] ✉, Jacky W. Y. Lam[2], Yi Zhang[1], Ben Zhong Tang[2,4] ✉ & Zhenguo Chi[1] ✉

Developing dynamic organic ultralong room-temperature phosphorescent (URTP) materials is of practical importance in various applications but remains a challenge due to the difficulty in manipulating aggregate structures. Herein, we report a dish-like molecular architecture via a bottom-up way, featuring guest-responsive dynamic URTP. Through controlling local fragment motions in the molecular architecture, fascinating dynamic URTP performances can be achieved in response to reversible accommodation of various guests, including solvents, alkyl bromides and even carbon dioxide. Large-scale regulations of phosphorescence lifetime (100-fold) and intensity (10-fold) can be realized, presenting a maximum phosphorescence efficiency and lifetime of 78.8% and 483.1 ms, respectively. Moreover, such a dish-like molecular architecture is employed for temperature-dependent multiple information encryption and visual identification of linear alkyl bromides. This work can not only deepen our understanding to construct multifunctional organic aggregates, but also facilitate the design of high-performance dynamic URTP materials and enrich their practical applications.

Organic ultralong room-temperature phosphorescence (URTP), generated from the radiative transition of triplet excitons in organic substance, is a persistent luminescence phenomenon after removal of the excitation source at room temperature[1–3]. Due to the unique photophysical property of long-lived emission lifetimes[4–6], as well as inherent merits of mild preparation conditions, facile modification and environmental friendliness[7–10], organic URTP materials are quite attractive for optoelectronics[11–13], molecular imaging[14–16], and many other fields[17–19]. In order to develop organic URTP materials with high efficiency and long lifetime, boosting the intersystem crossing (ISC) between singlet-triplet states and suppressing the nonradiative transition of triplet excitons in organic molecules are quite essential[20]. Following such basic principles, a lot of effort involving molecule engineering or two-component system fabrication, has gone into

[1]PCFM Lab, GDHPPC Lab, Guangdong Engineering Technology Research Center for High-performance Organic and Polymer Photoelectric Functional Films, State Key Laboratory of OEMT, School of Chemistry, Sun Yat-sen University, 510275 Guangzhou, China. [2]Department of Chemistry, Hong Kong Branch of Chinese National Engineering Research Center for Tissue Restoration and Reconstruction and Institute for Advanced Study, The Hong Kong University of Science and Technology, Kowloon, 999077 Hong Kong, China. [3]School of Materials Science and Engineering, Sun Yat-sen University, 510275 Guangzhou, China. [4]School of Science and Engineering, Shenzhen Institute of Aggregate Science and Technology, The Chinese University of Hong Kong, Shenzhen, 518172 Shenzhen, China. [5]These authors contributed equally: Wenlang Li, Qiuyi Huang. ✉e-mail: zhaoj95@mail.sysu.edu.cn; tangbenz@cuhk.edu.cn; chizhg@mail.sysu.edu.cn

**Fig. 1 | Design of the aggregate structure with dynamic URTP feature.** The bottom-up design strategy for the dish-like molecular architecture constructed by organic building blocks through intermolecular interactions.

developing high-performance URTP materials[21–27]. Among them, of particular interest are those dynamic organic URTP materials, which can exhibit reversible and significant changes in URTP performances in response to external stimuli[28]. On account of their unique stimuli-responsive features, such as photoactivation and excitation dependence[5,29], dynamic URTP materials have become a type of intelligent materials, demonstrating great promise for multicolor display, advanced anticounterfeiting, and many other crucial applications[30,31]. However, at present, most researches on dynamic URTP materials have only presented the URTP response to the stimulus of light[32,33], leading to the scarcity in terms of the types of external stimuli that dynamic URTP materials can respond to[34], which greatly impedes the further development of dynamic URTP materials for enriched practical applications.

The performance of URTP is intimately related to the molecular structure and molecular arrangement of organic substance, especially to their aggregate structures[35]. Thus, a deep understanding of structure-property relationship and a rational design of aggregate structure are quite crucial for developing novel dynamic URTP materials. For example, when the motions of local fragments in the molecular architectures can be regulated upon external stimuli, the nonradiative decay rate ($k_{nr}$) of triplet excitons can be further controlled. As a result, regulation of URTP properties and even an on/off URTP switching can be eventually achieved. However, the driving forces for most assemblies of URTP molecules are intermolecular noncovalent interactions, such as hydrogen bonds, π–π stacking, van der Waals force, which feature poor directionality during assembly[36]. Therefore, even a tiny structural modification to the molecular building block can lead to a huge change in its molecular architecture due to the weak intermolecular interactions, bringing an uncertainty about its URTP performance, let alone its dynamic behaviors[37]. Thereby, seeking for suitable molecular units with room-temperature phosphorescence property, of which assembling behaviors can be almost maintained after structural modification, is a quite challenging but significant work. In this case, a bottom-up way can be followed to construct ideal molecular architectures with versatile optical performances, and the relationship between aggregate structures and dynamic URTP properties can be well

investigated, which can provide new clues to enrich the types and applications of existing dynamic URTP materials.

Here, we designed and synthesized a dish-like molecular architecture featuring guest-responsive dynamic URTP property through a bottom-up method. Through intermolecular C−H···O＝S interactions, antiparallel molecular chains are constructed, in which antiparallel organic building blocks are further interlocked. Small cavities with guest accommodation abilities are formed between flexible phenothiazine moieties, eventually establishing a dish-like molecular architecture. Through reversibly controlling the accommodation and removal behaviors of guest molecules, to the best of our knowledge, the first example of fascinating dynamic URTP property with guest response can be realized in this molecular architecture. When guest molecules are accommodated in the cavities, molecular vibrations can be effectively restrained, leading to the dramatical decrease in the nonradiative decay rate of triplet excitons, thereby generating the significant URTP phenomenon with an impressive a phosphorescence quantum yield up to 78.8% and a lifetime up to 483.1 ms under ambient conditions. Furthermore, we demonstrate the applications of this molecular architecture with the guest-responsive dynamic URTP property in multiple information encryption and decryption, as well as visual identification of linear alkyl bromides with alkyl chain length selectivity.

## Results

### Design strategy for the organic building block

The organic building block of this molecular architecture, namely 10-(4-((4-(9H-carbazol-9-yl)phenyl)sulfonyl)phenyl)−10H-phenothiazine, henceforth denoted as CP, was designed and synthesized (Supplementary Fig. 1), which has been carefully purified by column chromatography and recrystallization for three times. The high purity of CP molecules is verified by [1]H and [13]C nuclear magnetic resonance (NMR) spectroscopies, as well as the high-resolution mass spectrum (HRMS) and high-performance liquid chromatography (HPLC) (Supplementary Figs. 2–5). As a typical ultralong room-temperature phosphorescence (URTP) molecule[38], diphenyl sulfone serves as the backbone of the CP molecule. In addition to the electron-rich S and O atoms, the rigid interlocking crystal structure of diphenyl sulfone also contributes greatly to the generation of its URTP phenomenon. In the crystal structure of pure diphenyl sulfone, the diphenyl sulfone molecule shows a unique mansard conformation with a dihedral angle of 76.34° between two phenyl rings. Diphenyl sulfone molecules are linked through intermolecular C−H···O＝S hydrogen-bonding interactions with a H···O distance of 2.462 Å, resulting in antiparallel molecular chains, in which two antiparallel diphenyl sulfone molecules present an interlocking packing mode with further assistance of intermolecular π−π interactions (Fig. 1 and Supplementary Fig. 14). Molecular crystals of diphenyl sulfone present a phosphorescence emission peaking at 529 nm, along with a phosphorescence quantum yield ($\Phi_P$) of 2.5% and a lifetime ($\tau$) of 43.2 ms.

Interestingly, when introducing a triplet luminophore, namely the carbazole moiety, to the 4-position of diphenyl sulfone, such an interlocking aggregate feature is well maintained, which is even still unchanged when other kinds of moieties are simultaneously connected to the 4′-position of diphenyl sulfone. When the other substituent group at the 4′-position of diphenyl sulfone is changed from the H atom to the Br atom or the methyl group, in single-crystal structures of these three organic compounds (named CH, CBr, and CM, respectively), the same packing mode constructed by antiparallel building blocks can all be continuously observed through intermolecular C−H···O＝S interactions with H···O distances of 2.808–3.564 Å and intermolecular π−π interactions (Supplementary Figs. 15–17). In the case of CH, benefiting from the interlocking molecular architecture, a URTP property with a phosphorescence quantum yield of 1.5% and an increased lifetime of 360 ms is successfully

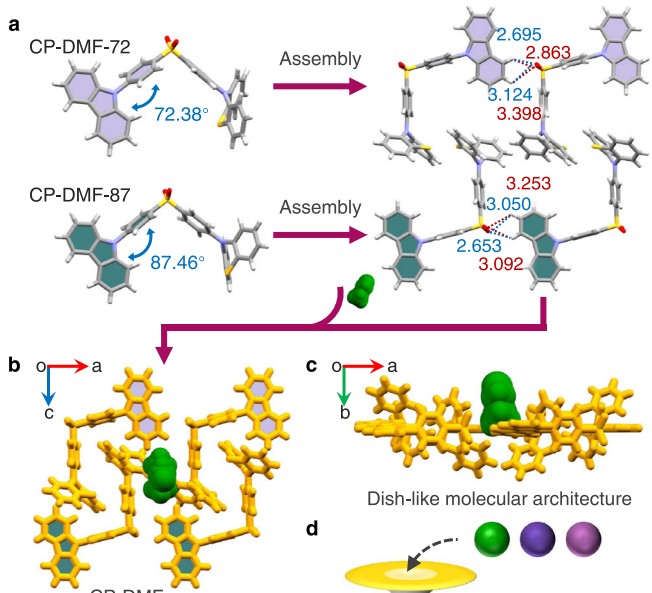

**Fig. 2 | Single-crystal structural investigation of CP-DMF. a** Two types of bimolecular units assembled from two CP-DMF-72 molecules (upper) and two CP-DMF-87 molecules (lower) in the single-crystal structure of CP-DMF. Color code: gray, C; red, O; blue, N; yellow, S; white, H. The dish-like molecular self-assembly architecture with a small cavity along **b** the *b*-axis and **c** the *c*-axis of the single-crystal structure, in which a DMF guest molecule is accommodated in its cavity. CP and DMF molecules are colored yellow and green, respectively. **d** Schematic diagram of the dish-like molecular architecture occupied by various guest molecules.

achieved (Fig. 1)[39]. Therefore, both the intermolecular interactions between organic building blocks and the distinct molecular conformations, play important roles in the constructions of such unique molecular architectures from various modified molecules. And by virtue of the interlocking molecular architecture based on the organic building block containing both diphenyl sulfone and carbazole moieties, the URTP property can be satisfactorily realized. Furthermore, although incorporating flexible organic moieties into the 4'-position of diphenyl sulfone, the performance of URTP can even be regulated, while the interlocking molecular architecture is still maintained. In that case, the flexible phenothiazine moiety is further incorporated, thereby producing the target building block of CP (Fig. 1). Interestingly, due to the folded conformation of the phenothiazine moiety, small cavities between phenothiazine moieties are observed in such a hydrogen-bonded organic framework, generating a dish-like molecular architecture, which can be occupied by various guests with certain sizes[36]. As a result, after inheriting the significant URTP property from the diphenyl sulfone backbone modified by the attachment of a carbazole moiety, the further introduction of a flexible phenothiazine moiety can probably endow the target building block and its eventual molecular architecture with extraordinary dynamic URTP phenomenon which can respond to various guest molecules.

## Single-crystal structure of the dish-like molecular architecture

Single crystals of the molecular architecture suitable for the single-crystal X-ray diffraction (SXRD) analysis, named CP-DMF, can be readily obtained by the vapor evaporation in a *N,N*-dimethylformamide (DMF) solution of CP molecules (Supplementary Fig. 18 and Supplementary Table 1). In the single-crystal structure of CP-DMF, two crystallographically independent CP molecules are observed, in which the dihedral angles between the plane of the carbazole moiety and the plane of the adjacent phenyl ring are 72.38° and 87.46°, respectively (Fig. 2a). The two types of CP molecules with different molecular conformations are further named CP-DMF-72 and CP-DMF-87,

respectively. Two neighboring CP-DMF-72 molecules are connected to each other through four C−H···O = S hydrogen-bonding interactions with H···O distances ranging from 2.695 to 3.398 Å, which also holds true for CP-DMF-87 molecules but with slightly different intermolecular C−H···O = S interaction distances. Along the *b*-axis of the single-crystal structure, two types of the bimolecular units assembled from two CP-DMF-72 molecules and two CP-DMF-87 molecules, respectively, are arranged antiparallel to each other, which are further interlocked (Supplementary Figs. 21–23 and Supplementary Tables 2, 3). Due to the distinct folded conformation of the phenothiazine moiety, a small cavity between two phenothiazine moieties contained respectively in CP-DMF-72 and CP-DMF-87 molecules is subsequently generated (Fig. 2b). As a result, along the *c*-axis, a dish-like molecular architecture with a small cavity can be successfully constructed by the self-assembly of four CP molecules, in which a DMF guest molecule is accommodated in its cavity (Fig. 2c, d). Notably, such dish-like molecular self-assembly architectures with small cavities can be continuously observed along the *c*-axis of the single-crystal structure (Supplementary Fig. 24).

## Dynamic URTP feature with reversible guest response

The photophysical properties of CP-DMF crystals with DMF accommodation are investigated under ambient conditions. Under irradiation by 365 nm UV light, the crystals exhibit strong green emission with an impressive photoluminescence (PL) quantum yield as high as 72.2%. Significantly, after turning off the UV-light source, the CP-DMF crystals show bright yellow green afterglow, which can be clearly observed by naked eyes even in the daylight condition (Fig. 3a, b). The spectrum profile delayed 8 ms of CP-DMF is quite similar to the steady-state one. From the lifetime decay profile, the yellow green emission is assigned to phosphorescence with a lifetime of 348.0 ms and a phosphorescence quantum yield of 60.7% under ambient conditions (Fig. 3d, Supplementary Fig. 25 and Supplementary Table 5)[40–42]. The phosphorescence nature of this ultralong emission can be further confirmed by the temperature-dependent PL spectra, as well as the temperature-dependent lifetime decay profiles. When the temperature is decreased, both the phosphorescence intensity and lifetime of CP-DMF crystals gradually increased (Supplementary Fig. 26 and Supplementary Table 4).

Interestingly, upon removing the DMF molecules in cavities, no afterglow can be observed by naked eyes in the resulting crystals of CP-Empty (Fig. 3b). According to the thermogravimetric analyses (TGA) data, DMF guest molecules in cavities of CP-DMF crystals can be released by heating at 80 °C, and the weight loss of 6% agrees well with the single-crystal structure of CP-DMF with DMF inclusion (Fig. 3c). Moreover, photophysical properties of CP-Empty crystals are in sharp contrast to that of CP-DMF crystals. The CP-empty crystals exhibit an extremely faded phosphorescence emission with a dramatically decreased lifetime of only 3.8 ms and a very low phosphorescence quantum yield of 4.8%, which are totally incapable of being caught by naked eyes (Fig. 3d and Supplementary Figs. 27–29). During the DMF removal process, as the powder X-ray diffraction (PXRD) patterns reveal, the desolvated crystal structure of CP-Empty is identical to its pristine solvated-type crystal structure of CP-DMF, while the crystallinity of CP-Empty can be retained without degradation of the crystal quality (Fig. 3e, f). Pawley refinements further determine the same molecular arrangement of CP-Empty and CP-DMF (Supplementary Figs. 30 and 31).

Remarkably, CP-DMF crystals can be facilely recovered from CP-Empty crystals upon exposure to DMF vapor, and then the DMF-regenerated CP-DMF crystals can be subsequently heated again to reproduce CP-Empty crystals (Fig. 3g). Such a cycling process can be repeated for several times without degradation of the URTP performance of CP-DMF crystals, simultaneously accompanied by a nearly one-hundred-time and ten-time regulations of phosphorescence

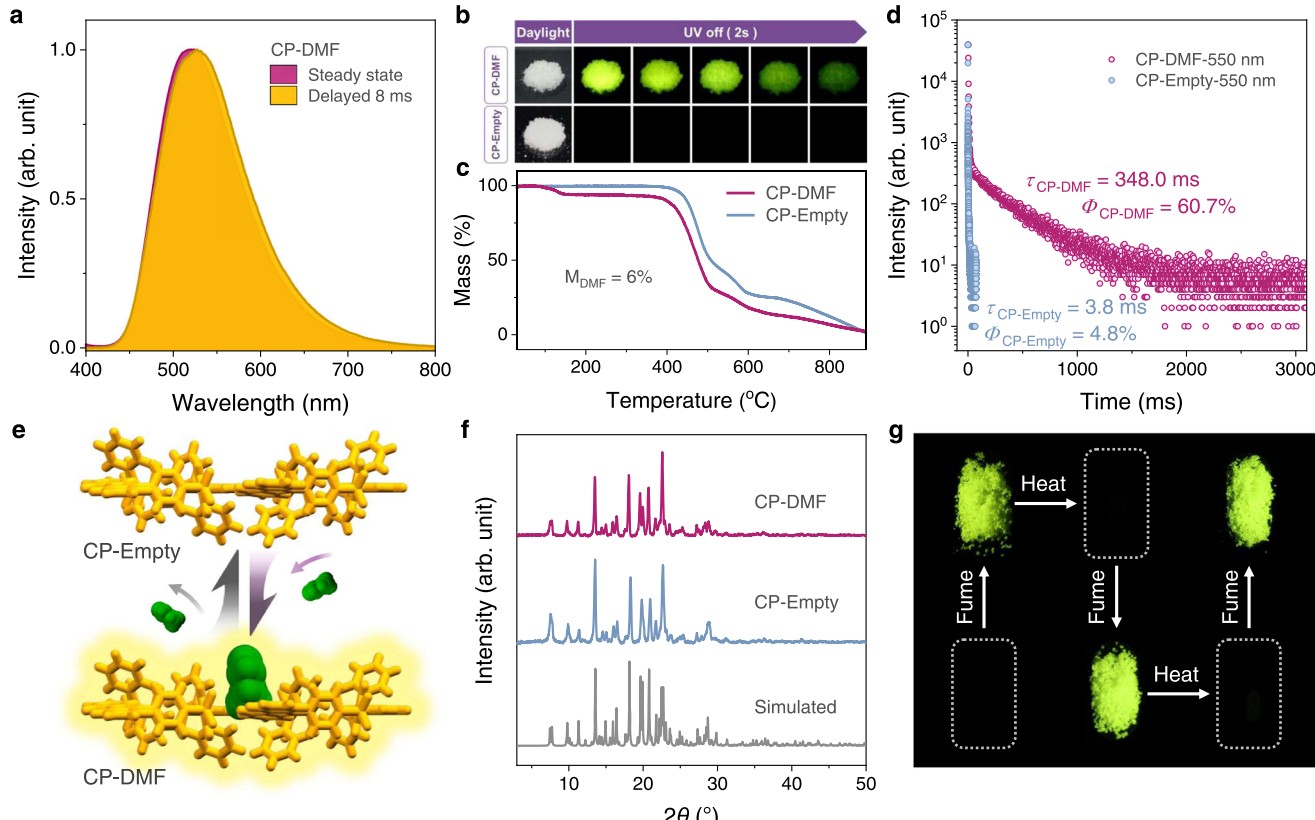

**Fig. 3 | Dynamic URTP feature characterization of CP-DMF. a** Steady photo-luminescence and phosphorescence spectra of CP-DMF crystals at room temperature under ambient conditions excited by 365 nm. **b** Photographs of CP-DMF and CP-Empty crystals taken in daylight and after turning off the UV-light source under ambient conditions. **c** TGA curves of CP-DMF and CP-Empty crystals. **d** Lifetime decay profiles of CP-DMF and CP-Empty crystals measured at 550 nm wavelength with a 355 nm spectral-LED source under ambient conditions. **e** Illustration of reversible guest accommodation and removal processes in the dish-like molecular architecture. CP and DMF molecules are colored yellow and green, respectively. **f** PXRD patterns of CP-DMF, CP-Empty, and the simulated PXRD pattern from single-crystal structure of CP-DMF. **g** Photographs presenting the dynamic URTP phenomenon of CP-DMF crystals in response to DMF accommodation and removal.

lifetime and intensity between CP-DMF and CP-Empty crystals, respectively. It is worth noting that, for human eyes, the phosphorescence lifetime of CP-Empty crystals is too short to be observed, and in other words, no URTP phenomenon of CP-Empty crystals can be observed by naked eyes. Therefore, though controlling the guest accommodation or removal processes in the dish-like molecular self-assembly architecture, a good reversibility between CP-DMF and CP-Empty crystals accompanied by the appearance and disappearance of URTP performances, namely the dynamic URTP property, are successfully realized. To the best of our knowledge, it is the first example of dynamic URTP phenomenon with the fascinating reversible guest response in such a molecular architecture.

**Mechanism of dynamic URTP emission**
Due to the existence of the small cavities (Fig. 4a), the dynamic URTP feature of this dish-like molecular architecture can probably be related to the high sensitivity of triplet excitons to oxygen or molecular motion. To gain deep insights into the mechanism of the guest-responsive dynamic URTP property, lifetime decay profiles of both CP-DMF and CP-Empty crystals are obtained under various conditions (Supplementary Figs. 32–34 and Supplementary Table 6). As clearly demonstrated in Fig. 4b, no matter under oxygen atmosphere or under vacuum, the lifetime of CP-Empty crystals is much shorter than that of CP-DMF crystals, which indicates that the dynamic URTP emission is not attributed to oxygen quenching. By contrast, once the crystals are cooled down from room temperature to 77 K, the lifetime of CP-Empty crystals becomes very close to that of CP-DMF crystals, demonstrating

that the thermal-induced molecular motion plays a decisive role in the dynamic URTP phenomenon of this molecular architecture. At very low temperature, molecular rotation and vibration can be strongly restrained, and thereby, the nonradiative decay rate ($k_{nr}$) of triplet excitons can be efficiently decreased, leading to the generation of long-lived phosphorescence. Similarly, when the small cavities are occupied by guest molecules, the free volume inside the crystal is greatly reduced, which can also effectively impede the molecular motion, especially the motion of flexible phenothiazine moieties, thus suppress the nonradiative decay to produce intense phosphorescence emission with long lifetime.

In order to provide further evidence for the molecular thermal motion in CP-Empty, carbon dioxide ($CO_2$) adsorption/desorption isotherms of CP-Empty crystals are recorded at various temperatures (Fig. 4c). Interestingly and abnormally, CP-Empty hardly adsorb $CO_2$ at 195 K, while raising the temperature to 273 K, the adsorption capacity is greatly increased compared to that at 195 K. And when the temperature is further increased to 298 K, a slight increase in $CO_2$ uptake can still be observed. The adsorbed amount of $CO_2$ at 298 K is five times greater than that at 195 K, which is opposite to the general trend of type I adsorbents presenting decreased gas uptake with elevated measurement temperature[43]. At 195 K, the vibration of the flexible phenothiazine moieties is restrained, which locks the diffusion pathways between cavities. Therefore, the cavities are almost isolated, which impedes the $CO_2$ adsorption. When raising the temperature, the thermal motion of flexible phenothiazine moieties is enhanced, and thereby, the cavity entrances become larger so that the $CO_2$ molecules

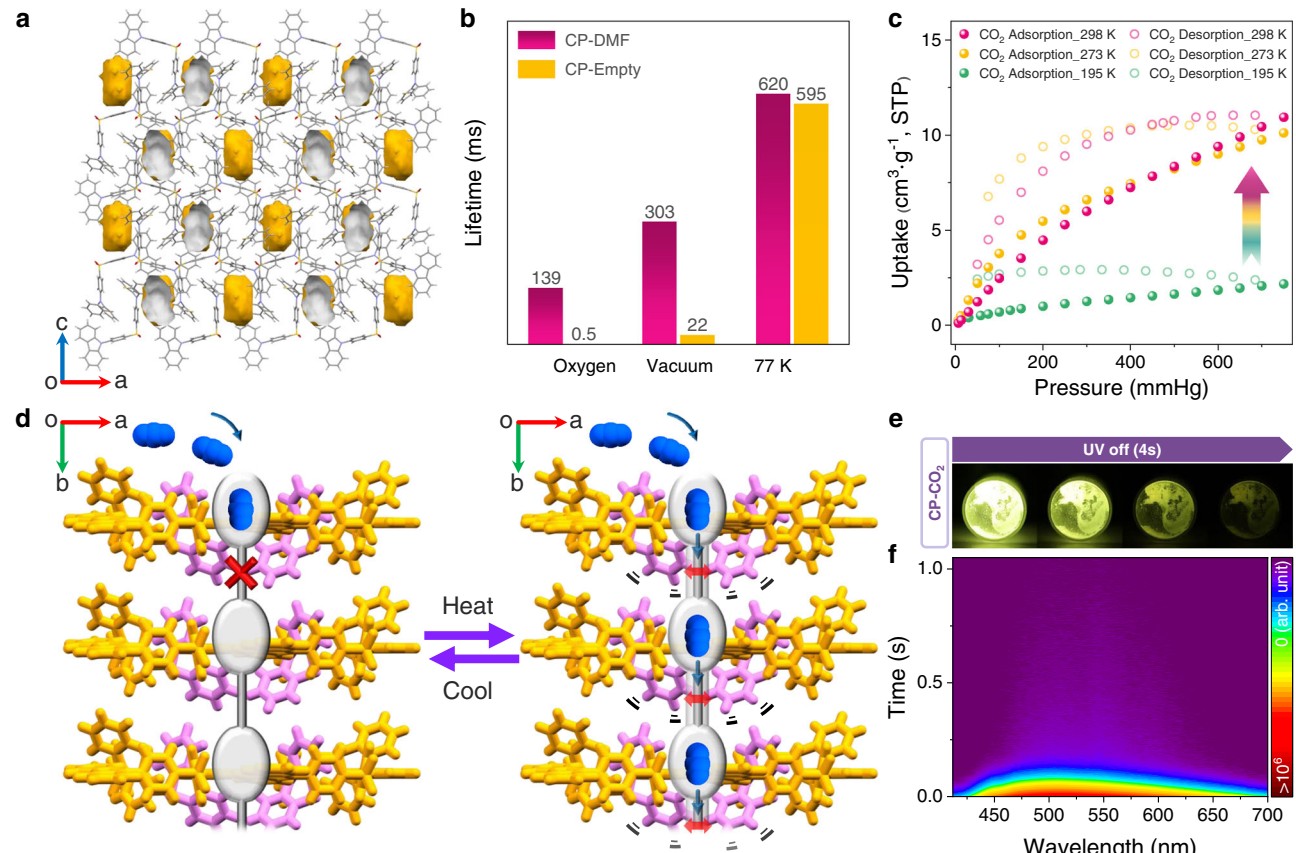

**Fig. 4 | Investigation of relationship between molecular motion and dynamic URTP feature. a** The single-crystal structure of CP-DMF along the *b*-axis, with the solvent-accessible void space visualized by gray/yellow (inner/outer) curved planes generated with a probe of 1.2 Å. Color code: gray, C; red, O; blue, N; yellow, S; white, H. DMF molecules are omitted. **b** Lifetimes of CP-DMF and CP-Empty crystals under various conditions. **c** $CO_2$ adsorption/desorption isotherms for CP-Empty at various temperatures. **d** Schematic diagram of void structures of the dish-like molecular architecture at low and high temperatures. Voids and $CO_2$ molecules are colored gray and blue, respectively. Flexible phenothiazine moieties and other parts of CP molecules are colored pink and yellow, respectively. **e** Photographs of the activated URTP phenomenon of CP-$CO_2$ crystals excited by 365 nm. **f** Time-resolved emission spectra of CP-$CO_2$ crystals with the excitation wavelength of 375 nm at room temperature under ambient conditions.

can diffuse into the open cavities (Fig. 4d)[44]. In particular, when $CO_2$ molecules are accommodated in voids of CP-Empty, resulting in crystals named CP-$CO_2$, a very strong URTP emission of CP-$CO_2$ crystals can also be activated and clearly observed by naked eyes under ambient conditions (Fig. 4e, f, Supplementary Fig. 35 and Supplementary Table 7), which further verifies that the molecular thermal motion is responsible for the dynamic URTP property. And remarkably, a dynamic URTP material in response to $CO_2$ gas molecules has been successfully achieved for the first time, which is quite promising for intelligent $CO_2$ sensing and detection.

As revealed in Fig. 5a, CP-DMF crystals exhibit dual phosphorescence under ambient conditions with phosphorescence peaks at 510 nm at a delay time of 8 ms and 550 nm at a delay time of 80 ms, which are ascribed to the radiative decays of excitons in a high-lying lowest triplet state ($T_1^H$) and a low-lying lowest triplet state ($T_1^L$), respectively (H/L represents high/low). According to the temperature-dependent lifetime decay profiles, when increasing the temperature from 100 to 300 K, the component ratio of $T_1^H$ excitons is continuously decreased while the component ratio of $T_1^L$ excitons is obviously increased (Fig. 5b, Supplementary Fig. 36 and Supplementary Table 8). These data suggest that the triplet excitons in the short-lived $T_1^H$ are effectively transferred to the long-lived $T_1^L$ through a triplet-triplet energy transfer (TTET) channel at elevated temperatures, thus producing the significant URTP phenomenon of CP-DMF crystals. To further uncover the TTET process, time-dependent density functional theory (TD-DFT) calculations based on the single-crystal structure of

CP-DMF are carried out (Supplementary Figs. 37, 38 and Supplementary Table 9). For two types of CP molecules with different molecular conformations in its single-crystal structure, namely CP-DMF-87 and CP-DMF-72, in their respective isolated single-molecular state, the lowest triplet states ($T_1$) are both local triplet ($^3$LE) states originating from the phenothiazine-substituted diphenyl sulfone moiety in CP-DMF-87 and the carbazole moiety in CP-DMF-72, respectively (Fig. 5c). In addition, although the calculated $T_1$ energy level of CP-DMF-87 is higher than that of CP-DMF-72, the spin-orbit coupling (SOC) constant between the ground state ($S_0$) and $T_1$ ($\xi_{SOT1} = 2.83$ cm$^{-1}$) of CP-DMF-87 is much larger than that of CP-DMF-72 ($\xi_{SOT1} = 0.23$ cm$^{-1}$), which implies that both of CP-DMF-87 and CP-DMF-72 can emit phosphorescence from their $T_1$ state, respectively. Accordingly, the $T_1^H$ and $T_1^L$ emissions of CP-DMF crystals peaking at 510 and 550 nm are dominated by the phenothiazine-substituted diphenyl sulfone moiety in CP-DMF-87 and the carbazole moiety in CP-DMF-72, respectively.

To further decipher this standpoint, two reference compounds, namely phenothiazine-disubstituted diphenyl sulfone and carbazole-disubstituted diphenyl sulfone (10,10'-(sulfonylbis(4,1-phenylene)) bis(10*H*-phenothiazine) and 9,9'-(sulfonylbis(4,1-phenylene))bis(9*H*-carbazole), named 2 P and 2 C, respectively) are synthesized (Supplementary Figs. 1, 6, and 7). In dilute dichloromethane solution at 77 K, two phosphorescence bands CP are identical with that of 2 P and 2 C, respectively, which indicates that different local fragments in CP are capable of emitting phosphorescence simultaneously (Supplementary Fig. 39). Therefore, we can propose a possible mechanism for the high-

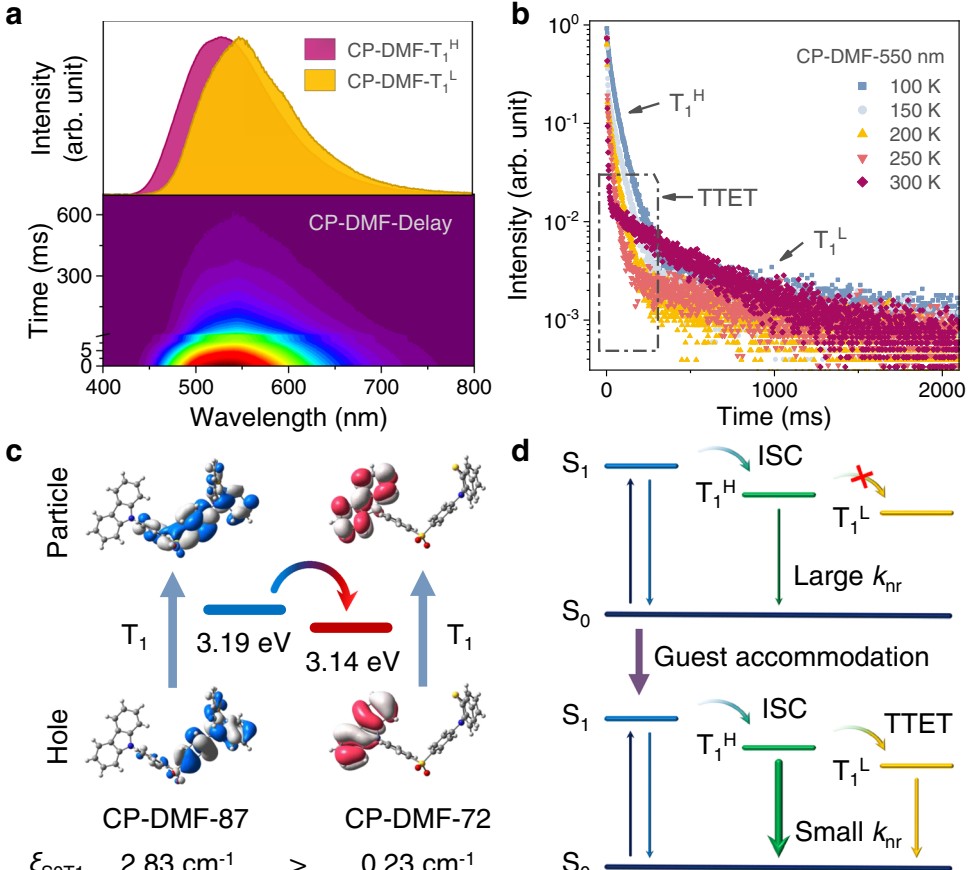

**Fig. 5 | Excited state process investigation of dynamic URTP feature. a** Time-resolved emission spectra of CP-DMF crystals under ambient conditions. The upper inset shows the phosphorescence spectra at delay times of 8 ms (pink) and 80 ms (yellow), respectively. **b** Temperature-dependent lifetime decay profiles of CP-DMF crystals measured at 550 nm wavelength with a 355 nm spectral-LED source under vacuum conditions. **c** Natural transition orbitals (NTOs) contributing to the lowest-energy triplet ($T_1$) transitions and spin-orbit coupling (SOC) constants between the ground state ($S_0$) and $T_1$ of single-molecular CP-DMF-87 and CP-DMF-72 in the single crystal of CP-DMF. **d** Proposed mechanism for the dynamic URTP property of the dish-like molecular architecture constructed by CP molecules before (upper) and after (lower) guest accommodation.

performance dynamic URTP property of such a disk-like molecular architecture based on CP molecules (Fig. 5d). In CP-Empty crystals, through the facile ISC process, the photoexcited excitons in the lowest singlet states ($S_1$) can be readily transferred to the $T_1^H$ dominated by the phenothiazine-substituted diphenyl sulfone moiety. However, due to the existence of empty cavities surround by flexible phenothiazine moieties, intense molecular thermal motion occurs in CP-Empty crystals. Consequently, the nonradiative transitions dominate the decays of triplet excitons in $T_1^H$. Even a very small fraction of the triplet excitons can be transferred to $T_1^L$ through the TTET channel, their nonradiative relaxation still play a dominant role, resulting in a poor phosphorescence performance. But when guest molecules are accommodated in these cavities, the molecular architecture, particularly the flexible phenothiazine moieties, can be stabilized by the steric effect and host-guest interactions. Nonradiative transitions can be suppressed to a great extent. Thus, most of relatively short-lived triplet excitons undergo radiative decays to produce the $T_1^H$ phosphorescence. And also, some of the triplet excitons can be further transferred to the carbazole-based $T_1^L$ through a TTET process, thus generating long-lived triplet excitons to produce the significant URTP emission of CP-DMF crystals. In a word, a fascinating dynamic URTP phenomenon with a nearly 100-fold regulation of phosphorescence lifetime and a 10-fold regulation of phosphorescence intensity can be achieved through guest accommodation/removal to manipulate the degree of non-radiative transitions in the dish-like molecular architecture.

The presence of isomeric impurities is recently reported to provide an alternative insight into the mechanisms behind the URTP property of carbazole derivatives[45,46]. In that case, positive absorption bands of radical ions generated by charge separation can be observed in transient absorption spectra. Nevertheless, for CP-DMF crystals herein from the commercial carbazole source, no transient absorption of charge-separated states (positive absorption) is observed, while only delayed emissions (negative absorption) are captured in transient absorption spectra (Supplementary Fig. 40). Therefore, the URTP property of CP-DMF crystals is resulted from the aforementioned ISC and TTET processes rather than charge-separated states from the isomer doping. In addition, we have synthesized CP molecules (named CP-Lab) from the synthesized carbazole. And we have also synthesized the target isomeric impurity (named CP-ISO) (Supplementary Figs. 1 and 8–13). The results show that, the URTP property of CP crystals is not generated from CP-ISO molecules, and the CP-Lab crystals also demonstrate the same dynamic URTP feature (Supplementary Figs. 41–46). Although the reversible guest accommodation/removal processes, the dynamic URTP phenomenon with the URTP performance ON/OFF switching can also be facilely realized in CP-Lab crystals, further suggesting that after stabilizing the flexible phenothiazine moieties though guest accommodation, nonradiative decays of triplet excitons can be effectively suppressed to produce the fascinating URTP emission in the molecular architecture.

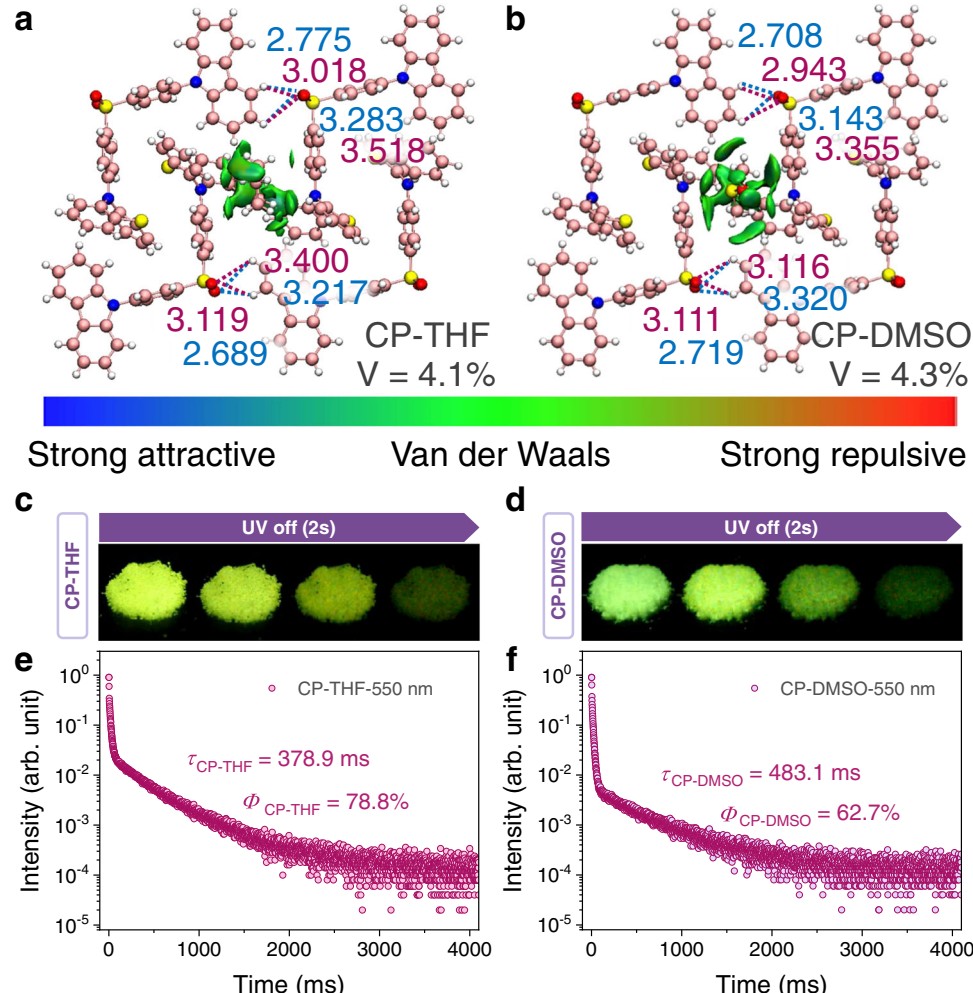

**Fig. 6 | Universality of dynamic URTP feature in response to various guests.** The distribution of intermolecular NCI regions at guest locations and intermolecular C–H···O = S interactions of single-crystal structures of **a** CP-THF and **b** CP-DMSO. Color code: pink, C; red, O; blue, N; yellow, S; white, H. Free volumes are generated with a probe of 0.7 Å. Photographs of URTP phenomenon of **c** CP-THF and **d** CP-DMSO crystals under ambient conditions excited by 365 nm. Lifetime decay profiles of **e** CP-THF and **f** CP-DMSO crystals measured at 550 nm wavelength with a 355 nm spectral-LED source under ambient conditions.

## Dynamic URTP feature for various guest accommodations

In addition to DMF, various solvent molecules including tetrahydrofuran (THF), dimethyl sulfoxide (DMSO) and dichloromethane (DCM) can all be accommodated in CP-Empty crystals, producing crystals of CP-THF, CP-DMSO, and CP-DCM. Molecular assembling forms of CP-THF, CP-DMSO, and CP-DCM are all consistent with that of CP-DMF (Supplementary Fig. 47), and significant URTP properties can all be observed in CP-THF, CP-DMSO, and CP-DCM crystals with quite similar URTP features to that of CP-DMF crystals (Fig. 6c–f and Supplementary Figs. 48–57).

Notably, single-crystal structures of CP-THF and CP-DMSO are also obtained, further revealing two crystallographically independent CP molecules and the same dish-like molecular self-assembly architectures as that of CP-DMF, in which THF and DMSO molecules are accommodated in their cavities, respectively (Fig. 6a, Supplementary Figs. 19, 20, 58–66 and Supplementary Tables 10–13). Intermolecular noncovalent interactions (NCI) between guests and architecture are visual clearly via the independent gradient model (IGM) analyses. Due to the various free volumes in their crystal structures (Supplementary Figs. 67–69), CP-DMF, CP-THF and CP-DMSO present different nonradiative decay rates, leading to slight differences in their URTP performances (Supplementary Table 5). A smallest free volume of 4.1% in CP-THF results in the outstanding URTP property with an even higher phosphorescence quantum yield of 78.8% and lifetime of 378.9 ms,

which is in accordance with the above-proposed mechanism of vibration-restraining induced dynamic URTP emission. The phosphorescence quantum yields of CP-DCM and CP-DMSO are 47.5% and 62.7%, while the lifetimes of them are 305.6 ms and 483 ms, respectively. As a result, the regulation of URTP performance can also be realized in this molecular architecture through various guest accommodations. Moreover, through controlling the accommodation and removal processes of these guest molecules, dynamic URTP properties can all be realized between CP-Empty crystals and various solvent-inclusion crystals respectively. The solvent escape temperatures for crystals with various solvent inclusions are quite different, which are dependent on the boiling points and vapor pressures of different solvents, as well as the intermolecular interactions in their crystal structures (Supplementary Fig. 70).

## Applications of dynamic URTP feature

In light of the interesting dynamic URTP feature of this molecular architecture, it is a good candidate for multiple information encryption and decryption. As shown in Fig. 7a, the original Chinese character "Shen" (means statement, etc.) is prepared by five kinds of CP crystals, which presents extremely strong green emission under 365 nm UV irradiation. After turning off the UV lamp, as the part made of CP-Empty crystals presents no URTP property, an encrypted Chinese character "Jia" (means the first, etc.) with yellow green URTP emission

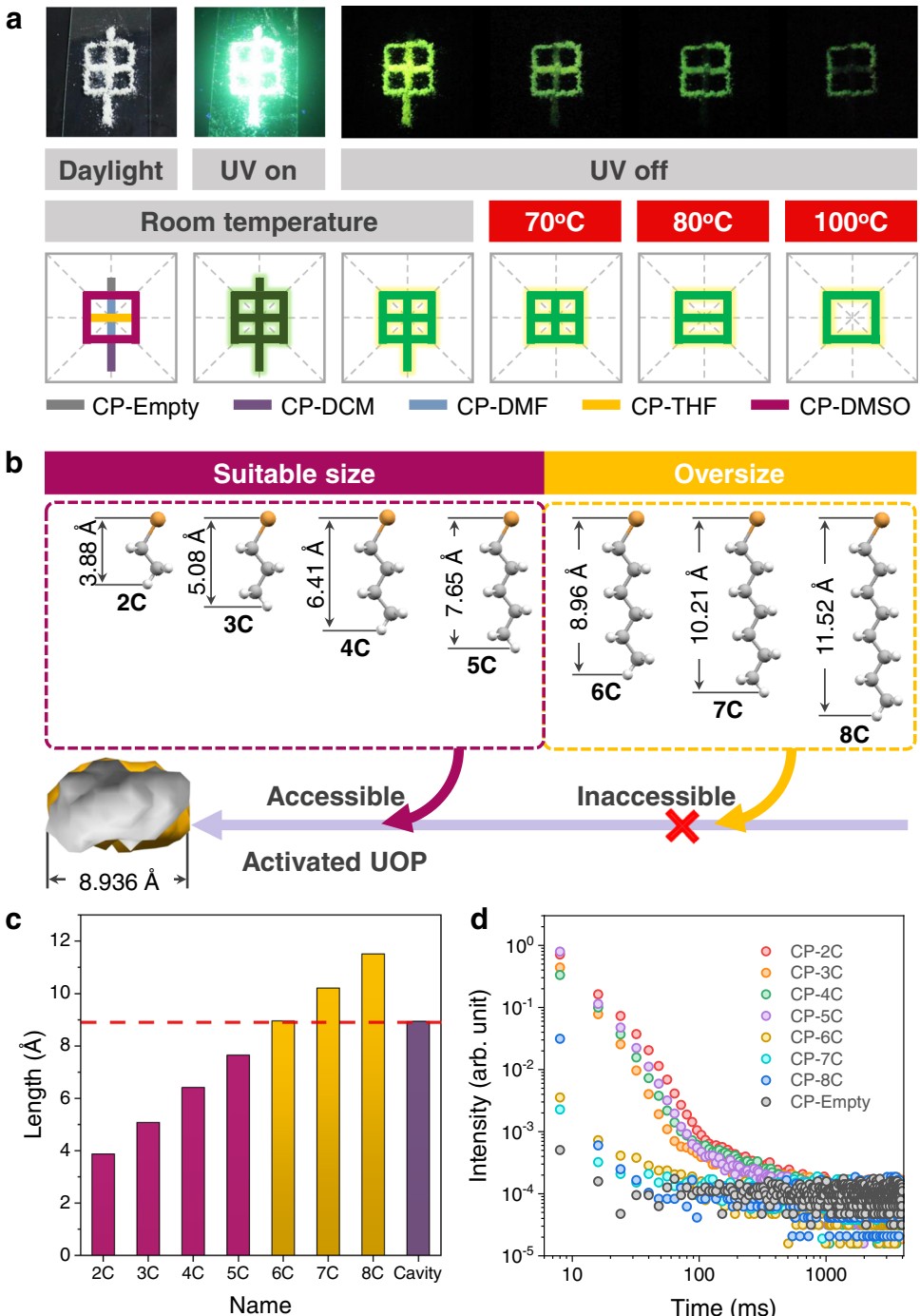

**Fig. 7 | Demonstration of dynamic URTP for temperature-dependent multiple information encryption and visual identification of linear alkyl bromides.** **a** Photographs of Chinese characters made of five kinds of CP crystals taken after different treatments under ambient conditions (upper) and illustration of the application in multiple information encryption and decryption based on the dynamic URTP feature of this molecular architecture (lower). **b** Optimized molecular structures of a series of linear alkyl bromides with various molecular dimensions, as well as the visualization of the cavity in the molecular architecture. **c** Molecular dimensions of linear alkyl bromides and the maximum dimension of window size in this molecular architecture. **d** Lifetime decay profiles of CP-Empty crystals and crystals after various alkyl bromide accommodations under ambient conditions.

is observed by naked eyes, showing a completely different meaning compared with the original one. Subsequently, when removing various guest molecules by heating at various temperatures, other three totally different encrypted Chinese characters "Tian" (means farmland, etc.), "Ri" (means sun, etc.), and "Kou" (means mouth, etc.) with yellow green URTP emissions can be visualized after switching off the UV source, respectively. Therefore, through a clever combination of various types of CP crystals, temperature-dependent multiple information

encryption and decryption has been successfully achieved on the basis of the dynamic URTP behavior of this molecular architecture.

In addition, given that the dynamic URTP character is achieved though controlling the guest behaviors in its cavities, the potential application of this molecular architecture in visual identification of linear alkyl bromides (named XC, X = 2–8, where X = the number of carbon atoms in its alkyl chain) with alkyl chain length selectivity is also demonstrated. Interestingly, the maximum dimension of window size

in this molecular architecture (8.936 Å) is larger than the molecular dimensions of alkyl bromides from 2 C to 5 C (ranging from 3.88 to 7.65 Å, Fig. 7b, c). Accordingly, 2C–5C can all be accommodated in the cavities of CP-Empty crystals, thereby activating the visual URTP phenomenon with long lifetimes (Fig. 7d and Supplementary Figs. 71–73). By contrast, alkyl bromides from 6 C to 8 C possess larger molecular dimensions ranging from 8.96 to 11.52 Å, making them hardly be accommodated in these cavities. Only very weak phosphorescence emissions with greatly decreased lifetimes are obtained, which could not be observed by naked eyes. As a result, cavities with appropriate size also render this molecular architecture a potential smart material, which exhibits dynamic URTP features in response to linear alkyl bromides with alkyl chain length selectivity.

## Conclusions

In summary, we have reported a dish-like molecular architecture featuring guest-responsive dynamic ultralong room-temperature phosphorescence (URTP) via a bottom-up method. Through intermolecular $C–H\cdots O=S$ interactions, antiparallel molecular chains are constructed, in which antiparallel organic building blocks are further interlocked. Small cavities are formed between flexible phenothiazine moieties, eventually establishing a dish-like molecular architecture. Through reversibly controlling the accommodation and removal behaviors of guest molecules, the first example of a molecular architecture exhibiting the dynamic URTP property with fascinating guest response can be realized. Various guests including different solvents, alkyl bromides and even carbon dioxide, can all induce the dynamic URTP performance of this dish-like molecular architecture with a good reversibility. When the cavities are occupied by guest molecules, molecular motion can be effectively restrained by the steric effect and host-guest interactions. Nonradiative decay of triplet excitons can thus be efficiently suppressed, leading to the significant URTP phenomenon in this dish-like molecular architecture with a highest phosphorescence quantum yield up to 78.8% and a longest lifetime up to 483.1 ms under ambient conditions. This work not only provides a valuable strategy to design novel dynamic URTP materials, but also offers an opportunity to develop a convenient platform for multiple information encryption and decryption, visual molecular identification, intelligent $CO_2$ sensing and detection, temperature-dependent display, advanced anticounterfeiting, and potentially many others.

## Methods

### Materials

Bis(p-fluorophenyl)sulfone, 10H-phenothiazine, carbazole, alkyl bromides, and sodium hydride (NaH) were purchased from J&K Scientific. 1-Bromo-2-nitrobenzene and phenylboronic acid were purchased from Energy Chemical. 1H-benzo[f]indole was purchased from TCI. *N,N*-dimethylformamide (DMF), dichloromethane (DCM), tetrahydrofuran (THF), dimethyl sulfoxide (DMSO), 1,2-dichlorobenzene (o-DCB), *n*-hexane (nH), and ethanol (EtOH) were all purchased as analytical grade from Guangzhou Dongzheng Co. (China).

### Measurement

$^1H$ and $^{13}C$ nuclear magnetic resonance (NMR) spectra were measured on a Bruker AVANCE III spectrometer in DMSO-$d_6$, with tetramethylsilane (TMS; $\delta = 0$) as the internal standard. High-resolution mass spectra (HRMS) were obtained on a GCT Premier CAB 048 mass spectrometer. High-performance liquid chromatography (HPLC) spectrum was measured on an Agilent Technologies 1260 Infinity with a column of Poroshell 120 (the eluting solvent was acetonitrile and water, flow rate = 0.2 mL/min). Photoluminescence and phosphorescence spectra were measured using an Ocean Optics QE65 Pro with a 365 nm Rhinospectrum RhinoLED as the excitation source or a Horiba Scientific Fluorolog-3 spectrofluorometer. The lifetime profiles and time-resolved

emission spectra were carried out with a Horiba Scientific Fluorolog-3 spectrofluorometer. The luminescence quantum yields were determined by using a Horiba Scientific Fluorolog-3 spectrofluorometer equipped with a Horiba Scientific Quanta-$\varphi$ calibrated integrating sphere. Transient absorption spectra were obtained in air using an Ocean Optics QE65 Pro with a 365 nm Rhinospectrum RhinoLED as the excitation source and an Ocean Optics DH-2000-BAL as UV-Vis-NIR light source. The photoluminescence images were taken by the Canon 750D digital camera. Temperature-dependent $CO_2$ gas adsorption/desorption isotherms were obtained by a Micromeritics ASAP 2020 surface area analyzer. The thermogravimetric analyses (TGA) data were obtained on a Shimadzu TGA-50 thermogravimetric analyzer. Powder X-ray diffraction (PXRD) data were collected using a Rigaku X-ray diffractometer (D/max-2200). The single-crystal X-ray diffraction (SXRD) data of single crystals were collected from an Agilent Technologies Gemini A Ultra system with Cu-Kα radiation. The structures were resolved and refined using direct methods with OLEX2. Solvent-accessible void spaces (free volumes) were calculated by mercury.

### Cultivation of single crystals

The single crystals of CP-DMF, CP-THF, and CP-DMSO suitable for the SXRD analysis were grown by solvent evaporation of CP solution in DMF, THF, and DMSO, respectively, at room temperature for 3 days. The single-crystal quality of CP-DCM is not good enough for the SXRD analysis because the solvent molecules escaped easily from crystals, leading to disordered phenothiazine moiety and poor-quality X-ray diffraction patterns.

### Data availability

Crystallographic data for all the different structures of CP reported in this article have been deposited at the Cambridge Crystallographic Data Centre (CCDC) under deposition numbers CCDC 2181479, CP-DMF; CCDC 2181480, CP-THF; CCDC 2181481, CP-DMSO; CCDC 2211685, CP-2C. The single-crystal structures of diphenyl sulfone, CH, CBr, and CM are obtained from previous papers. CCDC 1905615, diphenyl sulfone; CCDC 1402470, CH; CCDC 1402468, CBr; CCDC 1833075, CM. These data files can be obtained free of charge from www.ccdc.cam.ac.uk/data_request/cif. All other relevant data are available in the Supplementary Information.

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

## Acknowledgements

This work was financially supported by the National Natural Science Foundation of China (NSFC: 51733010, 61605253, 51973239) and the Fundamental Research Funds for the Central Universities. We thank Prof. Ian D. Williams and Dr. Herman H. Y. Sung in the Hong Kong University of Science and Technology for assistance on the SXRD measurement and analysis of CP-2C.

## Author contributions

W.L., Q.H., and Z.C. conceived and designed the experiments. W.L., Z.M., and X.H. performed the experiments. D.M., J.W.Y.L., and Y.Z. and all

other authors contributed to the data analyses. W.L. and Q.H. wrote the paper with the help of J.Z., Z.C., and B.Z.T. W.L. and Q.H. contributed equally to this work.

## Competing interests

The authors declare no competing interests.
