## [Peer Review File · Nature Communications]

A Dish-Like Molecular Architecture for Dynamic Ultralong Room-Temperature Phosphorescence through Reversible Guest AccommodationReviewers' Comments:

Reviewer #1:

Remarks to the Author:

In this work, the authors reported a dish-like molecular architecture with dynamic URTP, responding to reversible accommodation of various guests, and enabling large-scale regulations of phosphorescence lifetime and intensity. This work provides a simple and effective strategy to design novel dynamic URTP materials. Also, this paper is well written and sufficient data and analysis are provided. Therefore, I think it is worthy of publication in the journal after minor revisions, as below:

1. The author state that the emission peak at 510 nm can be assigned to phosphorescence. However, in the temperature-dependent PL spectra of CP-DMF and CP-Empty, the intensity of shoulder peaks at 470 nm is increased with the temperature increasing. please explain why.
2. It is a very interesting phenomenon that only one RTP peak can be observed in CP-Empty and two RTP peaks in CP-DMF, CP-THF and CP-DMSO can be observed. How about that in CP-CO₂? It seems there is only one RTP peak. Why?
3. Although the author has got the single-crystal structures of CP-THF and CP-DMSO, the simulated PXRD of both crystals should be given and compared to that of the CP-DMF for a clear view.
4. In Supplementary Figure 41a and 41b, the legends have small errors I think. It should be "1%CP-ISO/CP-Lab".
5. ¹H and ¹³C nuclear magnetic resonance spectra of CP-Lab should be given.

Reviewer #2:

Remarks to the Author:

Reviewer comment

In this manuscript, Li and co-workers describe a dish-like molecular architecture via a bottom-up way, featuring guest-responsive dynamic URTP. They have shown that dynamic URTP performances can be achieved in response to reversible accommodation of various guests, including solvents, alkyl bromides, and even carbon dioxide. Further, they claimed that large-scale regulations of phosphorescence lifetime (100-fold, 483.1 ms; quantum yield, 78.8%) and intensity (10-fold) could be realized. Finally, the author has used such dish-like molecular architecture for temperature-dependent multiple information encryption and visual identification of linear alkyl bromides by controlling local fragment motions in the molecular architecture. Overall, the detailed investigation of the crystalline compounds does conclude the observation, and the experimental results are carefully discussed with the obtained data. This result does bring urgency and significance to the field. This reviewer finds the novelty and urgency of this work to be published in this journal. Therefore, I would recommend that the authors submit their manuscript to this journal after the authors consider and address the following points.

1. Line 21: "phosphorescence" should be phosphorescent.
2. Line 113: "mansard roof conformation" should be mansard conformation to get proper proofreading.
3. Line 115 & 130: "C-H...O=S hydrogen-bonding interactions"- distance should be added.
4. In this manuscript, the author crystallizes all the materials in the presence of different solvents (guest) separately to achieve host-guest complexes. Surprisingly, the author did not show guest uptake in the amorphous state of the compounds. It would be more interesting if the author could show URTP via selective guest uptake.
5. The author has reported an exceptionally very high phosphorescence quantum yield (78.8%) with a high lifetime of 483 ms. The present reviewer finds it difficult to understand how such a high phosphorescence quantum yield can be possible with such a high lifetime, as the radiative rate is inversely proportional to the lifetime. The author should clearly explain this inconsistency. Further, the author should add a table of quantum yield and lifetime data, including fluorescence and phosphorescence process, which will bring more visibility to the readers. I would also like to see the

author has separated the fluorescence and phosphorescence quantum yields. These papers can be cited for the purpose. For example, *J. Phys. Chem. C.* 2022, 126, 5649–5657.; *J. Phys. Chem. Lett.*, 2018, 9, 2733-2738.; *Phys. Chem. Lett.*, 2018, 9, 3808-3813.

6. In HPLC analysis, the author should experiment for a long retention time to understand impurity, as impurity is a concern while claiming URTP.

7. Line 198: "phosphorescence" what is the delay used? The author should mention the delay in the text and make consistency throughout the manuscript.

8. Line 296: Figure 4b. It would be better to keep the decay plot representing the lifetime change under various conditions.

9. Line 308-314: what is the origin of two triplet states? The author must address the issue and make it consistent throughout the manuscript. I would suggest adding PL spectra to identify the source of the states.

10. It would be interesting if the author could produce crystals of compounds without solvent molecules to understand the cavity effect.

11. Line 399-401: The author must mention the quantum yields and lifetimes of the other host-guest systems for better visibility.

12. Line 436 & 444: "is made of five kinds of CP" & "readily visualized respectively" The author should check grammar issues throughout the manuscript.

13. The author should discuss crystals in the presence of alkyl halide if they have been produced. The crystals with alkyl halide may give a more interesting observation of the cavity effect.

14. Supplementary figure 65: Intensity (unnormalized) and a lifetime of CP-empty should be plotted together for better comparison. I suggest the author add unnormalized data to get better comparison results (if used for comparison).

Reviewers' comments:

Reviewer #1 (Remarks to the Author):

In this work, the authors reported a dish-like molecular architecture with dynamic URTP, responding to reversible accommodation of various guests, and enabling large-scale regulations of phosphorescence lifetime and intensity. This work provides a simple and effective strategy to design novel dynamic URTP materials. Also, this paper is well written and sufficient data and analysis are provided. Therefore, I think it is worthy of publication in the journal after minor revisions, as below:

1. The author state that the emission peak at 510 nm can be assigned to phosphorescence. However, in the temperature-dependent PL spectra of CP-DMF and CP-Empty, the intensity of shoulder peaks at 470 nm is increased with the temperature increasing. please explain why.
2. It is a very interesting phenomenon that only one RTP peak can be observed in CP-Empty and two RTP peaks in CP-DMF, CP-THF and CP-DMSO can be observed. How about that in CP-CO₂? It seems there is only one RTP peak. Why?
3. Although the author has got the single-crystal structures of CP-THF and CP-DMSO, the simulated PXRD of both crystals should be given and compared to that of the CP-DMF for a clear view.
4. In Supplementary Figure 41a and 41b, the legends have small errors I think. It should be "1%CP-ISO/CP-Lab".
5. 1H and 13C nuclear magnetic resonance spectra of CP-Lab should be given.

Reviewer #2 (Remarks to the Author):

Reviewer comment

In this manuscript, Li and co-workers describe a dish-like molecular architecture via a bottom-up way, featuring guest-responsive dynamic URTP. They have shown that dynamic URTP performances can be achieved in response to reversible accommodation of various guests, including solvents, alkyl bromides, and even carbon dioxide. Further, they claimed that large-scale regulations of phosphorescence lifetime (100-fold, 483.1 ms; quantum yield, 78.8%) and intensity (10-fold) could be realized. Finally, the author has used such dish-like molecular architecture for temperature-dependent multiple information encryption and visual identification of linear alkyl bromides by controlling local fragment motions in the molecular architecture. Overall, the detailed investigation of the crystalline compounds does conclude the observation, and the experimental results are carefully discussed with the obtained data. This result does bring urgency and significance to the field. This reviewer finds the novelty and urgency of this work to be published in this journal. Therefore, I would recommend that the authors submit their manuscript to this journal after the authors consider and address the following points.

1. Line 21: "phosphorescence" should be phosphorescent.
2. Line 113: "mansard roof conformation" should be mansard conformation to get proper proofreading.
3. Line 115 & 130: "C-H...O=S hydrogen-bonding interactions"- distance should be added.
4. In this manuscript, the author crystallizes all the materials in the presence of different solvents (guest) separately to achieve host-guest complexes. Surprisingly, the author did not show guest uptake in the

amorphous state of the compounds. It would be more interesting if the author could show URTP via selective guest uptake.

5. The author has reported an exceptionally very high phosphorescence quantum yield (78.8%) with a high lifetime of 483 ms. The present reviewer finds it difficult to understand how such a high phosphorescence quantum yield can be possible with such a high lifetime, as the radiative rate is inversely proportional to the lifetime. The author should clearly explain this inconsistency. Further, the author should add a table of quantum yield and lifetime data, including fluorescence and phosphorescence process, which will bring more visibility to the readers. I would also like to see the author has separated the fluorescence and phosphorescence quantum yields. These papers can be cited for the purpose. For example, J. Phys. Chem. C. 2022, 126, 5649–5657.; J. Phys. Chem. Lett., 2018, 9, 2733-2738.; Phys. Chem. Lett., 2018, 9, 3808-3813.

6. In HPLC analysis, the author should experiment for a long retention time to understand impurity, as impurity is a concern while claiming URTP.

7. Line 198: “phosphorescence” what is the delay used? The author should mention the delay in the text and make consistency throughout the manuscript.

8. Line 296: Figure 4b. It would be better to keep the decay plot representing the lifetime change under various conditions.

9. Line 308-314: what is the origin of two triplet states? The author must address the issue and make it consistent throughout the manuscript. I would suggest adding PL spectra to identify the source of the states.

10. It would be interesting if the author could produce crystals of compounds without solvent molecules to understand the cavity effect.

11. Line 399-401: The author must mention the quantum yields and lifetimes of the other host-guest systems for better visibility.

12. Line 436 & 444: “is made of five kinds of CP” & “readily visualized respectively” The author should check grammar issues throughout the manuscript.

13. The author should discuss crystals in the presence of alkyl halide if they have been produced. The crystals with alkyl halide may give a more interesting observation of the cavity effect.

14. Supplementary figure 65: Intensity (unnormalized) and a lifetime of CP-empty should be plotted together for better comparison. I suggest the author add unnormalized data to get better comparison results (if used for comparison).

Point-by-point response to the reviewers' comments:

(Reviewers' comments and suggestions: in black; Responses to the comments and suggestions: in blue)

Reviewer #1 (Remarks to the Author):

In this work, the authors reported a dish-like molecular architecture with dynamic URTP, responding to reversible accommodation of various guests, and enabling large-scale regulations of phosphorescence lifetime and intensity. This work provides a simple and effective strategy to design novel dynamic URTP materials. Also, this paper is well written and sufficient data and analysis are provided. Therefore, I think it is worthy of publication in the journal after minor revisions, as below:

Response: We would like to thank you for your positive comments and valuable suggestions to improve our manuscript.

1. The author state that the emission peak at 510 nm can be assigned to phosphorescence. However, in the temperature-dependent PL spectra of CP-DMF and CP-Empty, the intensity of shoulder peaks at 470 nm is increased with the temperature increasing. please explain why.

Response: We would like to thank you very much for your valuable comments. Except for the predominant phosphorescent peaks at 510 nm of CP-DMF and CP-Empty, there are still small emission peaks at 470 nm of CP-DMF and CP-Empty with thermally activated delayed fluorescence (TADF) property. The feature of TADF is related to the enhancement of emission intensity with temperature increasing because relative high temperature can boost the reverse intersystem crossing (RISC) rate of excitons from triplet state to singlet state (*Nature* **2012**, 492, 234-238). The requirement for molecules to have TADF properties is a small energy gap (ΔE_{ST}) between the lowest excited singlet state (S_1) and the lowest excited triplet state (T_1) (usually $\Delta E_{ST} \leq 0.3$ eV). In this manuscript, according to the TD-DFT calculations based on single-crystal geometry, ΔE_{ST} of CP-DMF-87 is 0.27 eV, which is lower than 0.3 eV (Supplementary Figure 38). Thus, it is reasonable for CP-DMF to have TADF property. Given the same PXRD pattern of CP-Empty as that of CP-DMF, the molecular conformation of CP-Empty is quite similar to that of CP-DMF, which indicates the small ΔE_{ST} of molecules in CP-Empty. As a result, it is also possible for CP-Empty to have TADF property.

Our changes made to the revised manuscript are highlighted in yellow as follows:

On page S21 in the revised Supporting Information:

Supplementary Figure 26 | Temperature-dependent **a**, photoluminescence spectra and **b**, lifetime decay profiles of CP-DMF under vacuum conditions with 350 nm excitation wavelength and a 355 nm spectral-LED source. Note: the enhancement of shoulder peaks at 470 nm with the temperature increasing is due to the thermally activated delayed fluorescence (TADF) property of CP-DMF.⁵

On page S23 in the revised Supporting Information:

Supplementary Figure 29 | Temperature-dependent **a**, photoluminescence spectra and **b**, lifetime decay profiles of CP-Empty under vacuum conditions with 350 nm excitation wavelength and a 355 nm spectral-LED source. Note: the enhancement of shoulder peaks at 470 nm with the temperature increasing is due to the thermally activated delayed fluorescence (TADF) property of CP-Empty.⁵

2. It is a very interesting phenomenon that only one RTP peak can be observed in CP-Empty and two RTP peaks in CP-DMF, CP-THF and CP-DMSO can be observed. How about that in CP-CO₂? It seems there is only one RTP peak. Why?

Response: We would like to thank you very much for your valuable comments. The possible mechanism for only one RTP peak in CP-Empty and two RTP peaks in CP-DMF, CP-THF and CP-DMSO is proposed in Fig. 5d. The key factor to possess two RTP peaks is whether the TTET between two triplet states is effective. In other words, due to the existence of empty cavities surround by flexible phenothiazine moieties, intense molecular thermal motion occurs in CP-Empty crystals, leading to larger k_{nr} to decrease TTET rate. But when guest molecules are accommodated in these cavities, such as CP-DMF, CP-THF and CP-DMSO, k_{nr} can be suppressed to a great extent so that the TTET process can occur effectively. As for CP-CO₂, the photophysical process should be similar to that of CP-DMF, CP-THF and CP-DMSO. In fact, there is also two RTP peaks in CP-CO₂. As shown in Fig. 4f, time-resolved emission spectra of CP-CO₂ crystals show a red-shift emission as delayed time goes on, which is not very clear, considering the component proportion of low-lying lowest triplet state is much small. Maybe the intermolecular interactions of CO₂ and CP molecules are weaker than that of other solvent guests and CP molecules due to the smaller size of CO₂.

3. Although the author has got the single-crystal structures of CP-THF and CP-DMSO, the simulated PXRD of both crystals should be given and compared to that of the CP-DMF for a clear view.

Response: We would like to thank you very much for your valuable comments. The simulated PXRD of CP-THF, CP-DMSO and CP-DMF are given and compared for a clear view. The PXRD patterns of CP-THF, CP-DMSO and CP-DMF are quite similar, indicating the molecular packing of CP-THF, CP-DMSO and CP-DMF are the same.

Our changes made to the revised manuscript are highlighted in yellow as follows:

On page 21 in the revised manuscript:

in which THF and DMSO molecules are accommodated in their cavities, respectively (Fig. 6a, Supplementary Figs. 19, 20, 58-66 and Supplementary Table 10-13).

On page S43 in the revised Supporting Information:

Supplementary Figure 58 | Simulated PXRD patterns of CP-DMF, CP-THF, and CP-DMSO according to their single crystals.

4. In Supplementary Figure 41a and 41b, the legends have small errors I think. It should be "1%CP-ISO/CP-Lab".

Response: Thanks for your careful revision. Some errors have been corrected in the revised manuscript.

On page S36 in the revised Supporting Information:

5. ^1H and ^{13}C nuclear magnetic resonance spectra of CP-Lab should be given.

Response: We would like to thank you very much for your valuable comments. ^1H and ^{13}C nuclear magnetic resonance spectra of CP-Lab are given. Our changes made to the revised manuscript are highlighted in yellow as follows:

On page S4 in the revised Supporting Information:

10-(4-((4-(9*H*-carbazol-9-yl)phenyl)sulfonyl)phenyl)-10*H*-phenothiazine (CP-Lab)

Sodium hydride (0.14 g, 5.9 mmol) was added into a solution of Cz-Lab (0.29 g, 1.8 mmol) in DMF (20 mL). After the mixture was stirred under nitrogen atmosphere for 15 min, FP (0.50 g, 1.2 mmol) was added. Then the mixture was heated up to 70°C and stirred for 6 h. After the mixture was cooled down to room temperature, the crude product was purified by silica gel column chromatography with DCM/*n*-hexane (v/v = 1:1) as eluent. Compound CP-Lab was obtained as a white solid in 60% yield (0.8 g). ^1H NMR (500 MHz, $\text{DMSO-}d_6$) δ : 8.25 (d, $J=7.7$ Hz, 2H), 8.16 (d, $J=8.6$ Hz, 2H), 7.91 (dd, $J=8.8, 3.0$ Hz, 4H), 7.57 (d, $J=7.6$ Hz, 2H), 7.49 (t, $J=8.0$ Hz, 2H), 7.46–7.38 (m, 6H), 7.32 (t, $J=7.1$ Hz, 4H), 7.13 (d, $J=9.0$ Hz, 2H). ^{13}C NMR (126 MHz, $\text{DMSO-}d_6$) δ : 149.05, 141.02.

On page S11 in the revised Supporting Information:

Supplementary Figure 11 | ^1H NMR spectrum of CP-Lab in $\text{DMSO-}d_6$.

Supplementary Figure 12 | ^{13}C NMR spectrum of CP-Lab in $\text{DMSO-}d_6$.

Reviewer #2 (Remarks to the Author):

Reviewer comment

In this manuscript, Li and co-workers describe a dish-like molecular architecture via a bottom-up way, featuring guest-responsive dynamic URTP. They have shown that dynamic URTP performances can be achieved in response to reversible accommodation of various guests, including solvents, alkyl bromides, and even carbon dioxide. Further, they claimed that large-scale regulations of

phosphorescence lifetime (100-fold, 483.1 ms; quantum yield, 78.8%) and intensity (10-fold) could be realized. Finally, the author has used such dish-like molecular architecture for temperature-dependent multiple information encryption and visual identification of linear alkyl bromides by controlling local fragment motions in the molecular architecture. Overall, the detailed investigation of the crystalline compounds does conclude the observation, and the experimental results are carefully discussed with the obtained data. This result does bring urgency and significance to the field. This reviewer finds the novelty and urgency of this work to be published in this journal. Therefore, I would recommend that the authors submit their manuscript to this journal after the authors consider and address the following points.

Response: We would like to thank you very much for your careful reviewing and positive comments on our manuscript. We are very pleased to learn your recognition of our work.

1. Line 21: “phosphorescence” should be phosphorescent.

Response: Thanks for your careful revision. The correction has been done in the revised manuscript.

On page 1 in the revised manuscript:

Developing dynamic organic ultralong room-temperature phosphorescent (URTP) materials is of practical importance in various applications but remains a challenge due to the difficulty in manipulating aggregate structures.

2. Line 113: “mansard roof conformation” should be mansard conformation to get proper proofreading.

Response: Thanks for your careful revision. The correction has been done in the revised manuscript.

On page 6 in the revised manuscript:

In the crystal structure of pure diphenyl sulfone, the diphenyl sulfone molecule shows a unique mansard conformation with a dihedral angle of 76.34° between two phenyl rings.

3. Line 115 & 130: “C–H···O=S hydrogen-bonding interactions”- distance should be added.

Response: Thanks for your careful revision. The corrections have been done in the revised manuscript.

On page 6 in the revised manuscript:

Diphenyl sulfone molecules are linked through intermolecular C–H···O=S hydrogen-bonding interactions with a H···O distance of 2.462 Å, resulting in antiparallel molecular chains, in which two antiparallel diphenyl sulfone molecules present an interlocking packing mode with further assistance of intermolecular π – π interactions...

...the same packing mode constructed by antiparallel building blocks can all be continuously

observed through intermolecular C–H \cdots O=S interactions with H \cdots O distances of 2.808 Å to 3.564 Å and intermolecular π – π interactions...

4. In this manuscript, the author crystallizes all the materials in the presence of different solvents (guest) separately to achieve host-guest complexes. Surprisingly, the author did not show guest uptake in the amorphous state of the compounds. It would be more interesting if the author could show URTP via selective guest uptake.

Response: We would like to thank you very much for your valuable comments. Through grinding samples, we can break the order crystal structure and obtain amorphous state of CP (named CP-G), which can be verified by PXRD. Compared to CP crystals, the steady luminescence of CP-G shows a red-shift emission with a 520 nm peak and a much shorter lifetime of phosphorescence. After fumed with *N, N*-dimethylformamide (DMF), tetrahydrofuran (THF), dimethyl sulfoxide (DMSO), dichloromethane (DCM), acetone (ACE) and ethyl acetate (EA), respectively, all of CP-G samples completed the transformation from amorphous states to crystalline states, showing vivid URTP and long lifetimes again.

Our changes made to the revised manuscript are highlighted in yellow as follows:

On page 21 in the revised manuscript:

Molecular assembling forms of CP-THF, CP-DMSO and CP-DCM are all consistent with that of CP-DMF (Supplementary Fig. 47), and significant URTP properties can all be observed in CP-THF, CP-DMSO and CP-DCM crystals with quite similar URTP features to that of CP-DMF crystals (Fig. 6c-6f and Supplementary Figs. 48-57).

On page S42 in the revised Supporting Information:

Supplementary Figure 57 | **a**, steady (solid lines) and delayed (dash lines) photoluminescence spectra and **b**, PXRD of CP-G, CP-G-DMF, CP-G-THF, CP-G-DMSO, CP-G-DCM, CP-G-ACE and CP-G-EA, respectively. ACE: acetone. EA: ethyl acetate. For the delayed spectra, except the delay time of CP-G is 5 ms, delay times of other samples are 10 ms. The lifetime decay profiles of **c**, CP-G, **d**, CP-G-DMF, CP-G-THF, CP-G-DMSO, CP-G-DCM, CP-G-ACE and CP-G-EA, respectively. The excitation wavelength is fixed at 375 nm. Note: Through grinding samples, amorphous state of CP can be obtained (named CP-G), which can be verified by PXRD. Compared to CP crystals, the steady luminescence of CP-G shows a red-shift emission with a 520 nm peak and a much shorter lifetime of phosphorescence. After fumed with various solvents, respectively, all CP-G samples completed the transformation from amorphous states to crystalline states, showing vivid URTP and long lifetimes again.

5. The author has reported an exceptionally very high phosphorescence quantum yield (78.8%) with a high lifetime of 483 ms. The present reviewer finds it difficult to understand how such a high phosphorescence quantum yield can be possible with such a high lifetime, as the radiative rate is inversely proportional to the lifetime. The author should clearly explain this inconsistency. Further, the author should add a table of quantum yield and lifetime data, including fluorescence and phosphorescence process, which will bring more visibility to the readers. I would also like to see the author has separated the fluorescence and phosphorescence quantum yields. These papers can be cited for the purpose. For example, *J. Phys. Chem. C.* 2022, 126, 5649–5657.; *J. Phys. Chem.*

Response: We would like to thank you very much for your valuable comments. According to the formula 1 below, in the case of phosphorescence, the quantum yield and lifetime seem to be a contradiction (Φ_P , phosphorescence quantum yield. k_r^T , radiative transition rate of phosphorescence. k_{nr}^T , nonradiative transition rate of phosphorescence. Φ_{ISC} , intersystem crossing efficiency. Phosphorescence lifetime will decrease with the increase of k_r^T).

$$\Phi_P = \frac{k_r^T}{k_r^T + k_{nr}^T} \Phi_{ISC} \quad \text{formula 1}$$

However, if we can reduce k_{nr}^T and enhance Φ_{ISC} , it is possible to obtain remarkable phosphorescence property with high phosphorescence quantum yield and ultralong lifetime. For example, to suppress non-radiative transitions and facilitates exciton generation by constructing high-density ionic bonding to chromophores, Prof. Wei Huang reported an organic phosphorescence molecule with 96.5% phosphorescence efficiency and 184.91 ms lifetime (*Nat. Mater.* **2021**, *20*, 1539-1544). Through encapsulating chromophores into polymers for fabricating abundant intermolecular interactions to suppress non-radiative transitions, Prof. Zhongfu An reported a phosphor with a record and quantum efficiency and lifetime up to 50.0% and 3.16 s simultaneously in film under ambient conditions (*Nat. Commun.* **2022**, *13*, 4890). In this paper, through DMF accommodation, local fragment motions in the molecular architecture can be effectively suppressed by the steric effect and host-guest interactions. Thus, the k_{nr}^T of CP-DMF (31.9 s^{-1}) is much smaller than that of CP-Empty (251.8 s^{-1}). Besides, due to the electron-rich heteroatoms of phenothiazine moiety, which can provide more n orbitals to boost ISC rate, CP-DMF shows efficient Φ_{ISC} . And then, through a TTET process from phenothiazine moiety to carbazole moiety, proportion of $\pi \rightarrow \pi^*$ transition is increased, which is in favor of long-lived phosphorescence. This viewpoint can also be supported by the previous paper (*Nat. Commun.* **2019**, *10*, 1595).

A table of photophysical parameters of CP-Empty, CP-DMF, CP-THF, CP-DMSO and CP-DCM is attached in Supplementary Table 5, including fluorescence and phosphorescence quantum yields and their lifetimes.

Three papers recommended are also cited.

Supplementary Table 5 | Photophysical parameters of CP-Empty, CP-DMF, CP-THF, CP-DMSO and CP-DCM.

Sample	CP-Empty	CP-DMF	CP-THF	CP-DMSO	CP-DCM
Φ_{total} (%) ^a	11.2	72.2	85.1	75.5	51.6
τ_F (ns)	3.9	3.6	2.7	3.1	2.6
$\square \Phi_F$ (%)	6.4	11.5	6.3	12.8	4.1
τ_P (ms) / A (%) ^b	0.02 / 22.18 0.7 / 34.78	2.2 / 15.95 12.3 / 22.15	1.7 / 7.43 16.7 / 34.95	2.2 / 17.03 14.2 / 52.38	2.1 / 8.00 18.7 / 64.09
$\square \Phi_P$ (%) ^c	4.8	60.7	78.8	62.7	47.5
$k_{nr,P}$ (s^{-1}) ^a	251.8	31.9	12.7	26.2	28.0
k_{ISC} (s^{-1}) ^a	1.2×10^7	1.7×10^8	2.9×10^8	2.0×10^8	1.8×10^8

^a $\Phi_{\text{total}} = \Phi_F + \Phi_P$; $k_{nr,P} = (1 - \Phi_P) / \tau_P$; $k_{ISC} = \Phi_P / \tau_F$.

^b Determined from the fitting function of $I(t) = A_1 e^{-t/\tau_1} + A_2 e^{-t/\tau_2} + A_3 e^{-t/\tau_3}$ according to

phosphorescence decay profiles.

$^c \Phi_p$ were calculated by the phosphorescence component of relevant lifetimes.¹¹

On page 11 in the revised manuscript:

From the lifetime decay profile, the yellow green emission is assigned to phosphorescence with a lifetime of 348.0 ms and a phosphorescence quantum yield of 60.7% under ambient conditions (Fig. 3d, Supplementary Fig. 23 and Supplementary Table 5).⁴⁰⁻⁴²

On page 34 in the revised manuscript:

40. Dey, S. et al. Thermally activated delayed fluorescence and room-temperature phosphorescence in asymmetric phenoxazine-quinoline (D2-A) conjugates and dual electroluminescence. *J. Phys. Chem. C* 126, 5649–5657 (2022).

41. Bhattacharjee, I., Acharya, N., Bhatia, H. & Ray, D. Dual emission through thermally activated delayed fluorescence and room-temperature phosphorescence, and their thermal enhancement via solid-state structural change in a carbazole-quinoline conjugate. *J. Phys. Chem. Lett.* 9, 2733–2738 (2018).

42. Bhatia, H., Bhattacharjee, I. & Ray, D. Biluminescence via fluorescence and persistent phosphorescence in amorphous organic donor(D4)–acceptor(A) conjugates and application in data security protection. *Phys. Chem. Lett.* 9, 3808–3813 (2018).

The serial numbers of the references are also revised again with double-check.

6. In HPLC analysis, the author should experiment for a long retention time to understand impurity, as impurity is a concern while claiming URTP.

Response: We would like to thank you very much for your valuable comments. To obtain HPLC peak with a long retention time, we used different eluting solvents and decreased flow rate to 0.2 mL/min. Our changes made to the revised manuscript are highlighted in yellow as follows:

On page 28 in the revised manuscript:

High performance liquid chromatography (HPLC) spectrum was measured on an Agilent Technologies 1260 Infinity with a column of Poroshell 120 (the eluting solvent was acetonitrile and water, flow rate = 0.2 mL/min).

On page S7 in the revised Supporting Information:

Supplementary Figure 5 | HPLC spectra of the CP molecule with acetonitrile-water as eluent in ratios of 100/0 and 90/10 (v/v).

7. Line 198: “phosphorescence” what is the delay used? The author should mention the delay in the text and make consistency throughout the manuscript.

Response: We would like to thank you very much for your valuable comments. We have supplemented the delay time in the text and make consistency throughout the manuscript.

Our changes made to the revised manuscript are highlighted in yellow as follows:

On page 11 in the revised manuscript:

The spectrum profile delayed 8 ms of CP-DMF is quite similar to the steady-state one. From the lifetime decay profile, the yellow green emission is assigned to phosphorescence with a lifetime of 348.0 ms and a phosphorescence quantum yield of 60.7% under ambient conditions

On page S31 in the revised Supporting Information:

Supplementary Figure 39 | Phosphorescence spectra of 2P, 2C and CP in DCM solutions with 10 μ M recorded at 77 K excited by 365 nm. The delay times are 8 ms. Note: The phosphorescence band at 510 nm of CP is identical with that of 2P, further suggesting that the T_1^H with an emission peak at 510 nm in CP-DMF crystals is generated from the phenothiazine-substituted diphenyl sulfone moiety. However, the emission peak at 550 nm in CP-DMF crystals is undetectable in the solution, and as a result, the T_1^L emission peak at 550 nm from the carbazole moiety is regulated and stabilized by the molecular aggregation.^{12,13}

8. Line 296: Figure 4b. It would be better to keep the decay plot representing the lifetime change under various conditions.

Response: We would like to thank you very much for your valuable comments. Considering the limited space of Figures in the manuscript, it is not easy to take six decay plots into one (which is Fig. 4b) and make them clear understanding for general readers at the same time. Therefore, we think that it will be better to compare the lifetimes of CP-DMF and CP-Empty one by one at three

different conditions, using a histogram way. Meanwhile, the original PL spectra and decay plots are also provided in the Supporting Information: for some special readers who have great interests to know more details.

Supplementary Figure 32 | **a**, photoluminescence spectra and **b**, lifetime decay profiles of CP-Empty under oxygen and vacuum conditions with 350 nm excitation wavelength and a 355 nm spectral-LED source.

Supplementary Figure 33 | **a**, photoluminescence spectra and **b**, lifetime decay profiles of CP-DMF under oxygen and vacuum conditions with 350 nm excitation wavelength and a 355 nm spectral-LED source.

Supplementary Figure 34 | Lifetime decay profiles of **a**, CP-DMF and **b**, CP-Empty at 77 K under vacuum conditions with a 355 nm spectral-LED source.

9. Line 308-314: what is the origin of two triplet states? The author must address the issue and make it consistent throughout the manuscript. I would suggest adding PL spectra to identify the source of the states.

Response: We would like to thank you very much for your valuable comments. We are sorry for our unclear description about the origin of two triplet states in our manuscript. In fact, we have added phosphorescence spectra of 2P (phenothiazine-disubstituted diphenyl sulfone), 2C (carbazole-disubstituted diphenyl sulfone) and CP in DCM solutions in supplementary Figure 35 to identify the source of two triplet states. Phosphorescence spectrum of CP exhibits two emission peaks at 420 nm and 510 nm, respectively, indicating CP owns two triplet states in solution state. Two phosphorescence peaks of CP are identical with that of 2C and 2P, respectively, revealing both carbazole and phenothiazine fragments of CP have capability to emit phosphorescence. For CP-DMF crystals, the T_1^H with an emission peak at 510 nm is generated from the phenothiazine-substituted diphenyl sulfone fragment, while the T_1^L emission peak at 550 nm from the carbazole moiety is regulated and stabilized by the molecular aggregation, according to the previous works by Prof. Wei Huang and Prof. Bin Liu (*Nat. Mater.* **2015**, *14*, 685-690; *Angew. Chem. Int. Ed.* **2022**, *61*, e202200343).

Supplementary Figure 39 | Phosphorescence spectra of 2P, 2C and CP in DCM solutions with 10 μM recorded at 77 K excited by 365 nm. The delay times are 8 ms. Note: The phosphorescence band at 510 nm of CP is identical with that of 2P, further suggesting that the T_1^{H} with an emission peak at 510 nm in CP-DMF crystals is generated from the phenothiazine-substituted diphenyl sulfone moiety. However, the emission peak at 550 nm in CP-DMF crystals is undetectable in the solution, and as a result, the T_1^{L} emission peak at 550 nm from the carbazole moiety is regulated and stabilized by the molecular aggregation.^{12,13}

10. *It would be interesting if the author could produce crystals of compounds without solvent molecules to understand the cavity effect.*

Response: We would like to thank you very much for your valuable comments. We have tried to produce crystals of CP-Empty for many times. Unfortunately, due to the poor quality of crystals, they are not suitable for the single-crystal X-ray diffraction (SXRD) analysis. Thus, the crystal structure of CP-Empty is determined via PXRD analysis. The result resolved by Pawley refinements shows content convergence with a Rwp value of 12.89% and the $\text{Pna}2_1$ space group with unit-cell parameters of $a = 24.6711 \text{ \AA}$, $b = 9.2634 \text{ \AA}$ and $c = 22.1711 \text{ \AA}$, featuring a slight contraction compared to the initial cell parameters of CP-DMF ($a = 26.0834 \text{ \AA}$, $b = 9.76020 \text{ \AA}$ and $c = 23.7023 \text{ \AA}$). As shown in supplementary Figure 31, after removal of DMF, the dish-like molecular architecture is retained with shorter $\text{H}\cdots\text{O}$ distances ranging from 2.508 to 3.209 \AA . More importantly, as expected, the cavities surround by flexible phenothiazine moieties are also retained with the void space of 323.60 \AA^3 (6.4% of the total cell volume). Consequently, the flexible fragments of CP-Empty have enough free space for thermal motion, leading to larger nonradiative transitions and weak phosphorescence emission of CP-Empty.

Supplementary Figure 31 | The intermolecular C–H···O=S interactions and cavity of the dish-like molecular architecture in the crystal structure of CP-Empty. Color code: grey, C; red, O; blue, N; yellow, S; white, H.

Our changes made to the revised manuscript are highlighted in yellow as follows:

On page 12 in the revised manuscript:

During the DMF removal process, as the powder X-ray diffraction (PXRD) patterns reveal, the desolvated crystal structure of CP-Empty is identical to its pristine solvated-type crystal structure of CP-DMF, while the crystallinity of CP-Empty can be retained without degradation of the crystal quality (Fig. 3e, 3f). Pawley refinements further determine the same molecular arrangement of CP-Empty and CP-DMF (Supplementary Figs. 30 and 31).

On page S24 in the revised Supporting Information:

Supplementary Figure 30 | Structural determination of CP-Empty by Pawley refinements based on PXRD patterns. Note: The result resolved by Pawley refinements shows content convergence with Rwp value of 12.89% and the Pna2₁ space group with unit-cell parameters of a = 24.6711 Å,

$b = 9.2634 \text{ \AA}$ and $c = 22.1711 \text{ \AA}$, featuring a slight contraction compared to the initial cell parameters of CP-DMF ($a = 26.0834 \text{ \AA}$, $b = 9.76020 \text{ \AA}$ and $c = 23.7023 \text{ \AA}$).

Supplementary Figure 31 | The intermolecular C–H...O=S interactions and cavity of the dish-like molecular architecture in the crystal structure of CP-Empty. Color code: grey, C; red, O; blue, N; yellow, S; white, H.

11. Line 399-401: The author must mention the quantum yields and lifetimes of the other host-guest systems for better visibility.

Response: We would like to thank you very much for your valuable comments. We are sorry for our unclear description about the quantum yields and lifetimes of CP-DCM, CP-THF and CP-DMSO in our manuscript. The quantum yields and lifetimes are added in the revised manuscript.

On page 22 in the revised manuscript:

A smallest free volume of 4.1% in CP-THF results in the outstanding URTP property with an even higher phosphorescence quantum yield of 78.8% and lifetime of 378.9 ms, which is in accordance with the above-proposed mechanism of vibration-restraining induced dynamic URTP emission. The phosphorescence quantum yields of CP-DCM and CP-DMSO are 47.5% and 62.7%, while the lifetimes of them are 305.6 ms and 483 ms, respectively.

12. Line 436 & 444: “is made of five kinds of CP” & “readily visualized respectively” The author should check grammar issues throughout the manuscript.

Response: Thanks for your careful revision. The corrections have been done in the revised manuscript.

On page 23-24 in the revised manuscript:

As shown in Fig. 7a, the original Chinese character “Shen” (means statement, etc.) is prepared by five kinds of CP crystals, which presents extremely strong green emission under 365 nm UV irradiation.

Subsequently, when removing various guest molecules by heating at various temperatures, other three totally different encrypted Chinese characters “Tian” (means farmland, etc.), “Ri” (means sun, etc.) and “Kou” (means mouth, etc.) with yellow green URTP emissions can be visualized after switching off the UV source, respectively.

13. The author should discuss crystals in the presence of alkyl halide if they have been produced. The crystals with alkyl halide may give a more interesting observation of the cavity effect.

Response: We would like to thank you very much for your valuable comments. Fortunately, we obtained the single-crystal structure of CP containing ethyl bromide (CP-2C) (Supplementary Figure 73). The molecular arrangement of CP-2C is quite similar to that of other crystals in the manuscript, showing a dish-like molecular architecture. That means, the mechanism of URTP of CP-2C is also the restriction of molecular motion, especially the motion of flexible phenothiazine.

PXRD results (Supplementary Figure 72a) show that after fuming with linear alkyl bromides (named XC, X = 2 to 8, where X = the number of carbon atoms in its alkyl chain), the molecular arrangements are not changed, which are the same with CP-Empty. The TGA curves (Supplementary Figure 72b) indicate the cavity of CP-Empty can accommodate 2C to 5C, while 6C-8C can not be accommodated due to the larger size.

Because the SXRD measurement and analysis of CP-2C is carried out with the help of Prof. Ian D. Williams and Dr. Herman H. Y. Sung in the Hong Kong University of Science and Technology, we appreciate their help and thank them at Acknowledgements.

Our changes made to the revised manuscript are highlighted in yellow as follows:

On page 24 in the revised manuscript:

Accordingly, 2C-5C can all be accommodated in the cavities of CP-Empty crystals, thereby activating the visual URTP phenomenon with long lifetimes (Fig. 7d and Supplementary Figs. 71-73).

On page 29 in the revised manuscript:

Crystallographic data for all the different structures of CP reported in this article have been deposited at the Cambridge Crystallographic Data Centre (CCDC) under deposition numbers CCDC 2181479, CP-DMF; CCDC 2181480, CP-THF; CCDC 2181481, CP-DMSO; CCDC 2211685, CP-2C.

On page 35 in the revised manuscript:

We thank Prof. Ian D. Williams and Dr. Herman H. Y. Sung in the Hong Kong University of Science and Technology for assistance on the SXRD measurement and analysis of CP-2C.

On page S55-56 in the revised Supporting Information:

Supplementary Figure 72 | **a**, PXRD patterns of CP-XC (X = 2 to 8) and CP-Empty. **b**, TGA curves of CP-XC (X = 2 to 8). Note: The same PXRD patterns indicate the dish-like molecular architectures of CP are still maintained after fuming with 2C to 8C, respectively. The mass losses of CP-2C to CP-5C in low temperature area as shown in TGA curves reveal the accommodation of 2C to 5C in cavity of CP-Empty, while 6C to 8C can not be accommodated due to the larger size.

Supplementary Figure 73 | The intermolecular C-H...O=S interactions in the crystal structure of CP-2C. Color code: grey, C; red, O; blue, N; yellow, S; white, H; brown, Br.

14. Supplementary figure 65: Intensity (unnormalized) and a lifetime of CP-empty should be plotted together for better comparison. I suggest the author add unnormalized data to get better comparison results (if used for comparison).

Response: We would like to thank you very much for your valuable comments. We are sorry for our unclear description about the supplementary Figure 65 (now it is supplementary Figure 71) in our manuscript. Because the measurement conditions of phosphorescence spectra of CP containing various linear alkyl bromides can not be totally the same, such as the quality and size of samples, unnormalized intensity can not reflect the truth accurately. To better compare results of CP-XC (X = 2-8) and CP-Empty, we can use the lifetime decay profiles (Fig. 7d) of CP-XC (X = 2-8) and CP-Empty to analyze the relative intensity of them at different decay times. For examples, at the delayed time of 8 ms, CP-2C keeps 71% intensity, while CP-6C and CP-Empty only remain 3.5% and 0.05% intensity, respectively. At the delayed time of 48 ms, CP-2C, CP-6C and CP-Empty hold 1%, 0.02% and 0.006% intensity, respectively. Because the size of 2C is suitable to be adapted by the cavity of CP-Empty, the molecule motion of CP-Empty will be restricted, thus exhibiting effective RTP. Although the size of 6C is too large to be adapted by the cavity of CP-Empty, owing to the external heavy atom effect, the RTP performance of CP-6C is slightly better than that of CP-Empty. As for the supplementary Figure 71, it can also show the phosphorescence spectra profiles of CP-XC (X = 2-8) at various decay times.

Fig. 7d Lifetime decay profiles of CP-Empty crystals and crystals after various alkyl bromide accommodations under ambient conditions.

Considering that some new Supplementary Figures are added during the revision, the serial numbers of Supplementary Figures are also re-ordered in the revised manuscript and Supporting Information with double-check.

Reviewers' Comments:

Reviewer #1:

Remarks to the Author:

The authors have revised the manuscript carefully. Therefore, I recommend its publication as it is.

Reviewer #2:

Remarks to the Author:

my responses are well-addressed. Now this revised manuscript can be published.

Reviewers' comments:

Reviewer #1 (Remarks to the Author):

The authors have revised the manuscript carefully. Therefore, I recommend its publication as it is.

Reviewer #2 (Remarks to the Author):

my responses are well-addressed. Now this revised manuscript can be published.

Point-by-point response to the reviewers' comments:

(Reviewers' comments and suggestions: in black; Responses to the comments and suggestions: in blue)

Reviewer #1 (Remarks to the Author):

The authors have revised the manuscript carefully. Therefore, I recommend its publication as it is.

Response: We would like to thank you very much for your valuable comments on our manuscript.

Reviewer #2 (Remarks to the Author):

my responses are well-addressed. Now this revised manuscript can be published.

Response: We would like to thank you very much for your helpful suggestions and positive comments to polish our paper.